# Spatiotemporal Fine-grained Video Description for Short Videos

## ABSTRACT

In the mobile internet era, short videos are inundating people's lives. However, research on visual language models specifically designed for short videos has not yet received sufficient attention. Short videos are not just videos of limited duration. The prominent visual details and high information density of short videos differentiate them to long videos. In this paper, we propose the SpatioTemporal Fine-grained Description (STFVD) emphasizing on the uniqueness of short videos, which entails capturing the intricate details of the main subject and fine-grained movements. To this end, we create a comprehensive Short Video Advertisements Description (SVAD) dataset, comprising 34,930 clips from 5,046 videos. The dataset covers a range of topics, including 191 sub-industries, 649 popular products, and 470 trending games. Various efforts have been made in the data annotation process to ensure the inclusion of fine-grained spatiotemporal information, resulting in 34,930 high-quality annotations. Compared to existing datasets, samples in SVAD exhibit a superior text information density, suggesting that SVAD is more appropriate for the analysis of short videos. Based on the SVAD dataset, we develop a visual language model (SVAD-VLM) to generate spatiotemporal fine-grained description for short videos. We use a prompt-guided keyword generation task to efficiently learn key visual information. Moreover, we also utilize dual visual alignment to exploit the advantage of mixed-datasets training. Experiments on SVAD dataset demonstrate the challenge of STFVD and the competitive performance of proposed method compared to previous ones.

## CCS CONCEPTS

• **Computing methodologies** → **Video summarization**; • **Information systems** → **Multimedia databases**.

## KEYWORDS

Multimodal Large Language Model, Short Video Advertisements Description, Visual Language Model

## 1 INTRODUCTION

Short videos lasting less than one minute have become popular owing to the emergence of social media platforms such as Tik-Tok. However, they are not just characterized by limited duration. Short videos, featuring a quick-swipe, single-column format in apps, differ from long-form videos posted on platforms such as

YouTube by cutting down switching costs and ramping up competition. The differences in duration and dissemination method distinguish short videos from long-form videos. On one hand, compared to movies and TV shows, the visual details in short videos play an more important role in affecting the audience's perception. The high pixel density of small screens for short videos, combined with their simple composition, leads to clear images and facilitates easy absorption of visual details by viewers. On the other hand, short videos contain a high amount of information due to the short durations and limited user attention spans. With the low switching cost and intense competitions, short videos typically lack setting shots, with each shot aiming to present climax as much as possible. As a result, existing video based text generation tasks, such as video caption [6, 12, 35] and dense video caption [10, 11, 46], are not suitable for short videos.

To this end, we propose a new task called SpatioTemporal Fine-grained Video Description (STFVD). The aim of this task is to provide comprehensive descriptions of the video subject in both spatial and temporal dimensions. Wang et al. [33] reports that the incorporation of key elements, which we term as video subjects, can significantly enhance the effectiveness of advertising short videos. Spatially, STFVD requires comprehensive details about the subject because details in the short video are more prominent compared to longer formats, thus requiring a level of granularity often overlooked in standard video captioning tasks. Temporally, STFVD requires a sequential and comprehensive depiction of a series of movements throughout the video to effectively capture the highlights of short videos. This differs from conventional video captioning that might offer sparse and condensed summaries. In summary, focusing on short videos, STFVD provides a higher level of visual details and movements, resulting in higher text information density compared to video captioning and dense video captioning (Figure 2b). Existing datasets are inadequate for this task. Therefore, we create the Short Video Advertisements Description (SVAD) dataset. This dataset comprises 34,930 clips extracted from 5,046 short videos, covering a wide range of information relevant to advertising scenarios. This dataset spans over 191 sub-industries, 649 popular products, and 470 trending games. We establish a rigorous and detailed annotation protocol and provide comprehensive training to annotators to ensure the inclusion of fine-grained spatiotemporal information in the annotations, resulting in high-quality annotations with rich semantics and high linguistic complexity.

Despite assembling the SVAD dataset, we are faced with two significant challenges. The first challenge is related to the richness of semantics in STFVD annotations. In spatiotemporal fine-grained video descriptions, each annotation typically contains more nouns and verbs compared to those in video captioning (see Figure 2), which increases the difficulty in generating accurate results. The second challenge is the generalization capabilities of the model. Advertisement short videos comprise a wide range of categories and exhibit a high degree of diversity, thus training a model with robust generalization capabilities requires a large amount of data.

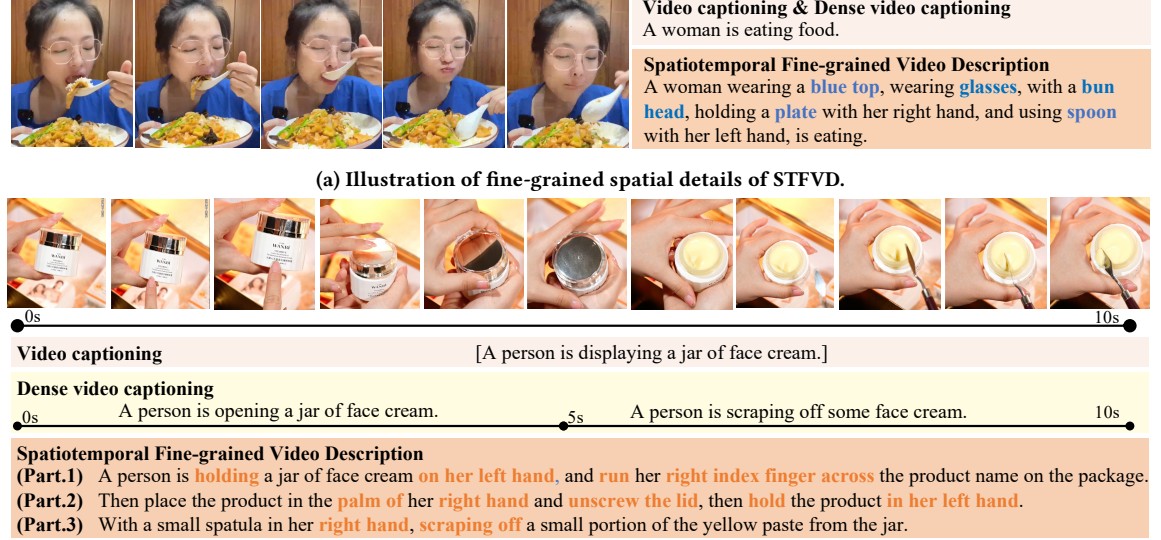

**Video captioning & Dense video captioning**
A woman is eating food.

**Spatiotemporal Fine-grained Video Description**
A woman wearing a **blue top**, wearing **glasses**, with a **bun head**, holding a **plate** with her right hand, and using **spoon** with her left hand, is eating.

**(a) Illustration of fine-grained spatial details of STFVD.**

0s																		10s

**Video captioning**										[A person is displaying a jar of face cream.]

**Dense video captioning**
0s				A person is opening a jar of face cream.						5s				A person is scraping off some face cream.						10s

**Spatiotemporal Fine-grained Video Description**
**(Part.1)** A person is **holding** a jar of face cream **on her left hand**, and **run** her **right index finger across** the product name on the package.
**(Part.2)** Then place the product in the **palm of** her **right hand** and **unscrew the lid**, then **hold** the product **in her left hand**.
**(Part.3)** With a small spatula in her **right hand**, **scraping off** a small portion of the yellow paste from the jar.

**(b) Illustration of fine-grained temporal details of STFVD.**

**Figure 1: The distinctions between video captioning, dense video captioning, and spatiotemporal fine-grained video description (STFVD). Visual details are highlighted in blue, whereas movement details are marked in orange. Figure 1a showcases the depth of STFVD on spatial (picture) level, offering a richer visual context (e.g., blue top, glasses, bun hairstyle) compared to the more general approaches of video captioning and dense video captioning. Figure 1b delves into the temporal (movement) level, where STFVD surpasses traditional captioning methods by providing detailed descriptions of movements, closely mirroring the video's content.**

However, providing high-quality annotations for such a large-scale dataset is not economically viable due to the high costs associated with manual labeling.

In this work, we propose Short Video Advertisements Description Visual Language Model (SVAD-VLM) to generate fine-grained descriptions for short video advertisements. To overcome the difficulty in generating rich semantic descriptions, we utilize prompt-guided keyword generation to enable the model to concentrate on key information within annotations, such as visual details and fine-grained movements of video subjects. Moreover, we investigate utilizing abundant video caption data to enhance the generalization capabilities of SVAD-VLM across diverse advertisement scenarios. In order to adapt different text patterns in various datasets, we propose dual visual alignment to boost the efficiency of mixed training with heterogeneously annotated datasets. We achieve state-of-the-art results on the SVAD dataset. Comprehensive experiments are conducted to demonstrate the effectiveness of our method.

The contributions of this paper are summarized as follows:

- We describe the uniqueness of short videos in video understanding and present a new problem of spatiotemporal fine-grained video description.
- We create the Short Video Advertisements Description (SVAD) dataset with videos from a broad spectrum of categories and spatiotemporal fine-grained descriptions of considerable linguistic complexity. SVAD is, to the best of our knowledge, the first dataset aimed at the fine-grained description of short video advertisements.
- We develop SVAD-VLM to facilitate spatiotemporal fine-grained video description for short video advertisements. We

develop prompt guided keyword generation to overcome the challenge posed by rich semantic fine-grained descriptions. Furthermore, we introduce dual visual alignment to leverage the benefits of mixed training with auxiliary datasets and enhance the model's generalization capability.

## 2 RELATED WORKS

**Video Captioning Datasets and Methods.** Video captioning aims to describe the main event and content in a video with a few simple and concise natural language sentences [6, 12, 17, 23, 29, 35, 37]. Furthermore, one video with multiple events can also be captured in dense video captioning task [10, 11, 46], which automatically localizes the multiple temporal events and then generates the descriptions one by one. However, video captioning and dense video captioning only provide simple description, which is not enough to understand and sort out the rich information in the short video era. The work most closely related to ours is FAVD [27]. However, SVAD dataset is made up of short videos collected from a main stream short video platform, which distinct from long-form videos by attracting visual details and high information density. There are datasets in the era of e-commerce [43, 44], but they only provide description of products [43] or video-related advertising slogan [44]. Instead, SVAD provides comprehensive video description including subjects and movements and covers a wide range of industries such as finance, healthcare and game (see Figure 3).

Recent video captioning methods [16, 34, 39] achieve impressing performance on video caption and dense video caption, but they are not suitable for STFVD because of the domain gap of short videos and long-form videos. Specifically, Lavender [16] also adopts

**Table 1: Statistical information in comparison with other datasets.**

| Dataset | Video | | Annotation | | Language |
|---|---|---|---|---|---|
| | #Clip | DUR(s) | #Word/s | #Word/Annotation | |
| MSVD [6] | 1,970 | 9.7 | 0.92 | 8.7 | English/Chinese |
| MSR-VTT [9] | 10,000 | 14.8 | 0.66 | 9.3 | English |
| VATEX [35] | 41,250 | 10.0 | 1.46 | 14.6 | English/Chinese |
| TVC [12] | 21,793 | 76.2 | 3.11 | 13.4 | English |
| YouCookII [46] | 15,433 | 41.0 | 0.82 | 7.9 | English |
| FAVDBench [27] | 11,424 | 7.7 | 5.54 | 17.4 | English/Chinese |
| Chinaopen-1k [5] | 1,092 | 32.5 | 0.44 | 14.2 | English/Chinese |
| SVAD | 34,930 | 5.1 | 12.2 | 45.2 | English/Chinese |

multi-task training. However, the purpose of this work is to unify different tasks at the expense of a slight sacrifice in accuracy. In contrast, our goal is to enhance the model's performance on the primary data as much as possible through carefully designed multi-task, multi-dataset training.

**Visual Language Models.** Recent advancements in the field of large visual language models have shown remarkable efficacy across a spectrum of visual language tasks. Pioneering models [1, 8, 24, 31] typically adopt a structured approach comprising a visual encoder, a visual adaptor, and a comprehensive large language model [4, 26]. These models undergo a pretraining phase on extensive visual language datasets, effectively integrating visual context into the LLM. In terms of the image language model, BLIP2 and InstructBLIP [7, 13] employ the Q-Former, a transformer visual adaptor, to extract language-aligned visual features. LLaVA [21] implements a straightforward projector for the integration of visual data into Vicuna [45], concurrently training the LLM during the fine-tuning phase. These methodologies underscore the efficacy of incorporating pretrained visual features into LLMs, significantly enhancing the model's ability to interpret and understand images In terms of video language model, InternVideo [36] is a general video foundation model, which combines masked video encoder VideoMAE [28] with spatial-temporal video model UniFormerv2 [15]. VideoChat [14] is proposed to design a multimodal dialogue system adopting a Q-Former architecture to integrate the video tokens from InternVideo with the large language model Vicuna. VideoLLaMA [42] combines Q-Former with the large language model LLaMA [30] and designs two branches to realize video language alignment and audio language alignment. Video-LLaVA [18] extents LLaVA to video understanding by inducing joint training of image and video.

## 3 SHORT VIDEO ADVERTISEMENTS DESCRIPTION DATASET

In this section, we present the Short Video Advertisements Description dataset from four aspects. In Section 3.1, we introduce Spatiotemporal Fine-grained Video Description task, focusing on uniqueness of short videos. In Section 3.2, we discuss collection and cleaning process of the dataset, highlighting a diverse range of high-quality video samples. Moving to Section 3.3, we describe our approach to data annotation. Spot-checking and self-review by annotators are employed to enhance the quality of labeling. In Section 3.4, we offer an exhaustive statistical overview of the dataset, demonstrating its broad diversity and differences to existing datasets.

**Table 2: Statistical information of dataset splits.**

| Split | Industry | | Entity | | Game | |
|---|---|---|---|---|---|---|
| | #Videos | #Clips | #Videos | #Clips | #Videos | #Clips |
| Train | 2,965 | 22,477 | 827 | 5,116 | 731 | 3,574 |
| Test | 376 | 2,968 | 60 | 361 | 87 | 464 |
| Total | 3,341 | 25,445 | 887 | 5,477 | 818 | 4,038 |

### 3.1 Spatiotemporal Fine-grained Video Description

The task of STFVD involves generating descriptions that are both spatially and temporally detailed for short video clips. This task requires the generation of descriptions that convey information about all subjects within the video, adhering to two primary rules. **Rule No.1**: Important visual details about the video's subjects must be described. Given that short videos are often viewed on relatively small phone screens and tend to have simpler compositions than films or TV shows, these visual details are crucial for capturing the audience's attention and thus are essential for inclusion in the description. **Rule No.2**: The movements of all subjects within the video should be described in sequence. Short videos are of higher information density than long videos due to limited length and competitions of attention spans of users, movement level instead of action level description can reduce the loss of information.

### 3.2 Data Collection

To ensure the dataset includes a broad spectrum of advertising short videos, a uniform sampling from an extensive range of sub-industries is conducted. With e-commerce and game identified as the leading industries in short video advertising, a deliberate selection of advertisements for trending products and popular games is made to enhance the dataset's diversity.

To streamline the annotation workflow, an in-house segmentation model is utilized to facilitate the annotation process by delineating individual shots within the videos. It is observed that the rapid pace of short videos frequently lead to a high occurrence of extremely brief segments. These segments, providing limited temporal context, are deemed suboptimal for the intended descriptive analysis within an advertising framework. As a remedy, a dual filtration strategy is implemented: videos with more than 30 percent of their duration consisting of short clips are classified as fragmented and excluded; furthermore, within each chosen video, brief clips shorter than a specific threshold are disregarded.

In our comprehensive sampling, a collection of 3,341 short videos is compiled, representing over 15 categories and 191 sub-industries. To capture a wide array of product diversity, an additional 887 videos are selected based on top product categories, showcasing 649 trending products. Recognizing the significant role of gaming in advertising, 818 gaming-related short videos from 470 renowned games are also included. The distribution and examples of videos across various industries, products, and games, depicted in Figure 3, demonstrate the substantial diversity within the Short Video Advertisements Description (SVAD) dataset.

This strategic curation process culminated in a dataset comprising 34,930 individual clips from 5,046 short videos.

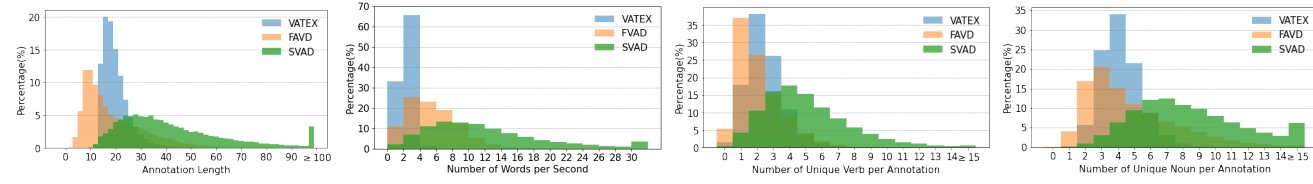

(a) Annotation lengths.  (b) Words per second.  (c) Unique verbs per annotation.  (d) Unique nouns per annotation.

**Figure 2: Statistical histogram distributions on VATEX [35], FAVD [27], and SVAD. Compared to VATEX and FAVD, distributions of SVAD shift to the right, which means its annotations are longer and richer in verbs and nouns, with a higher text information density.**

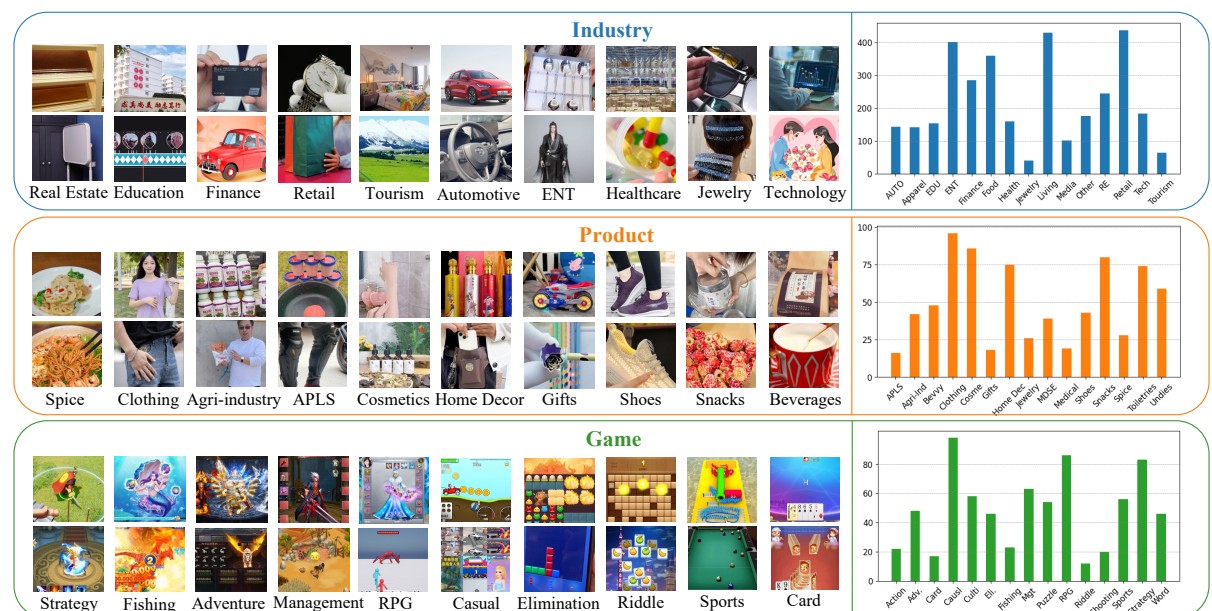

**Figure 3: The statistical histogram distribution and examples of SVAD videos in industry, product, and game categories. The height of the bars indicates the number of videos in each class. Video distribution across different industry categories is colored blue. The distributions of the top 15 product categories and the top 15 game categories are colored orange and green, respectively. This figure demonstrates that SVAD exhibits substantial diversity.**

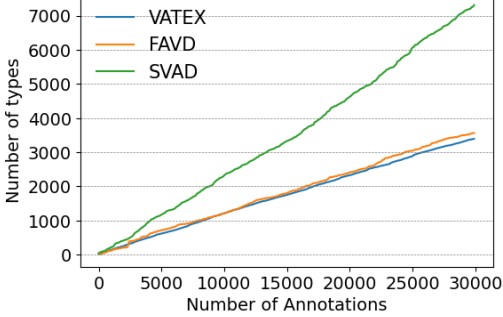

**Figure 4: Type-annotation curve, with "type" referring to a unique 4-gram. Our SVAD shows more annotation diversity than other datasets.**

### 3.3 Data Annotation

Expert annotators are hired by a professional annotation company to label the data. Comprehensive guidelines for annotation are crafted, and a range of example annotations are supplied. It is

discovered that beyond standard annotation guidelines, an abundance of illustrative examples significantly assists annotators in accurately understanding these standards, thereby enabling the production of high-quality annotations.

To stringently control the quality of annotations, daily quality checks are conducted at the end of each annotation session. Work from annotators who do not meet the quality criteria is subject to re-evaluation by the annotation team. Moreover, after the initial round of annotations, trained annotators are required to review their peers' work and make necessary corrections to any substandard annotations. These practices ensure the high annotation quality of the dataset. As a result, a total of 34,930 valid annotations are obtained.

### 3.4 Dataset Statistics

The compiled dataset encompasses 34,930 segments extracted from 5,046 videos, showcasing an extensive variety covering 649 different products, 470 various games, and 191 distinct sub-industries. This diversity is depicted in Figure 3. The dataset is segmented into

Table 3: POS tags and average unique *n*-grams.

| Dataset | Fine-grained | POS tag | | | | Average Unique N-grams | | | |
|---|---|---|---|---|---|---|---|---|---|
| | | % Adv. | % Verb | % Noun | % Adj. | 1-gram | 2-gram | 3-gram | 4-gram |
| MSVD | - | 3.7 | 17.9 | 33.5 | 2.0 | 0.4 | 1.7 | 2.7 | 2.9 |
| VATEX | - | 6.4 | 16.4 | 28.4 | 3.7 | 0.1 | 0.8 | 2.2 | 3.5 |
| FAVD | ✓ | 2.9 | 15.3 | 33.9 | 8.4 | 0.2 | 1.7 | 4.2 | 6.0 |
| SVAD | ✓ | 3.7 | 18.4 | 32.7 | 4.9 | 0.6 | 4.8 | 10.7 | 15.4 |

training and testing sets; the former comprises 4,523 short videos and 31,167 clips, while the latter includes 523 short videos with 3,793 segments. Detailed information regarding the dataset division is presented in Table 2. On average, each short video consists of seven clips, with an average duration of 5.1 seconds per clip. The annotations have an average length of 45.2 words.

In comparison with other datasets relevant to video captioning and dense video captioning, as shown in Table 1, we highlight our **text information density** by calculating the number of words per annotation divided by the duration of a video clip **(Word/s)**. It is observed that while many datasets score around 1, our SVAD dataset reaches the highest text information density of 12.2, doubling the highest value (5.5) found among competitors. This significant variance, attributed to the high information density characteristic of short videos and the fine-grained nature of STFVD, highlights the domain gap between short and conventional videos.

A more thorough comparison is conducted with the well-established video captioning dataset VATEX [35] and another fine-grained video description dataset, FAVD [27]. Compared to VATEX [35] and FAVD [27], all four distributions of SVAD shift to the right, as depicted in Figure 2, suggesting that each SVAD annotation is characterized by a greater length (Figure 2a), encompassing more nouns (Figure 2d) and verbs (Figure 2c). As demonstrated in Figure 2b, SVAD boasts a higher count of video-annotation pairs with elevated text information density in comparison to the other datasets.

To examine linguistic complexity, unique n-grams per annotation and Part-of-Speech (POS) are computed , as detailed in Table 3. A higher count of unique n-grams per annotation in SVAD suggests superior linguistic complexity, presenting a challenge for precise generation. The POS analysis reveals that datasets focused on fine-grained descriptions, such as FAVD [27] and SVAD, contain a higher percentage of adjectives than those centered on video captioning. Moreover, the proportion of verbs in SVAD exceeds that in FAVD [27] and other datasets, indicating that SVAD annotations feature more movements, primarily because short video shots often capture intense plot conflicts or highlights.

Inspired by the type-token vocabulary curve [41] and the Type-Caption Curve in VATEX [35], the Type-Annotation Curve is illustrated in Figure 4. Each data point $(x, y)$ on this curve signifies the count of unique types $y$ in a corpus with $x$ annotations, with SVAD showcasing the steepest curve. This implies that SVAD possesses a higher linguistic complexity compared to other datasets.

## 4 METHOD

Given an advertisement short video that is divided into multiple segments $\{x_1, ..., x_n\}$ , the objective is to generate a spatiotemporal fine-grained video description for each segment. Specifically, for the

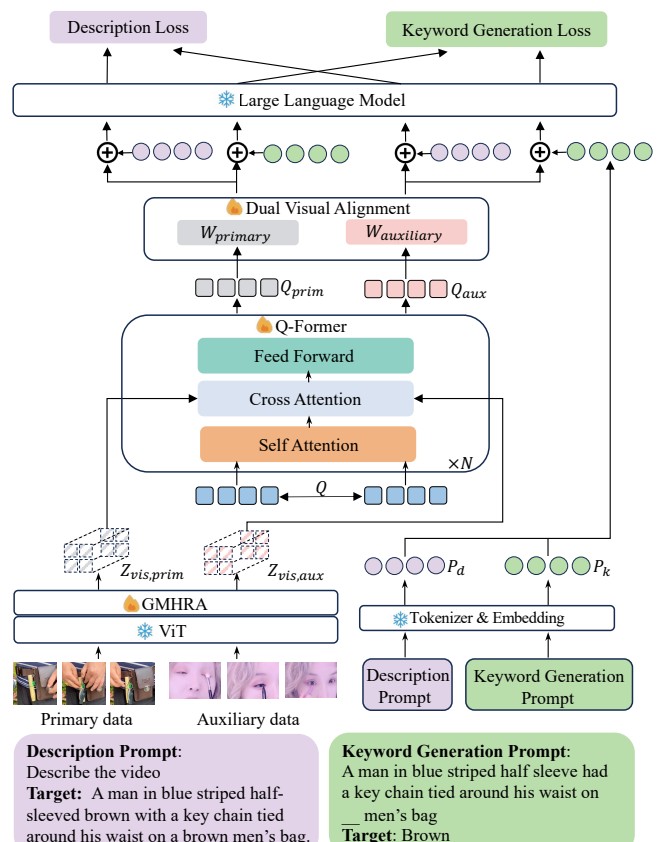

Figure 5: Overview of SVAD-VLM method. Data from various datasets are processed through a vision encoder and a Q-Former, after which they are projected into the embedding space of LLM by different components of the dual visual alignment layer. The learnable queries from primary and auxiliary data are concatenated with either a description prompt or a keyword generation prompt. The concatenated sequence is then fed into the LLM to produce response. Subsequently, these responses are optimized with the corresponding description loss or keyword generation loss. Examples of description prompt and keyword generation prompt are illustrated on bottom.

multiple frames $x_i = \{I_1, I_2, ...I_T\}$ of the i-th short video segment, the aim is to product a text that sequentially describes the subject's movements within the segment, while capturing relevant details. This section begins with an introduction to the model structure

(Section 4.1), proceeds to discuss the prompt-guided keyword generation task (Section 4.2), and concludes with the presentation of the dual visual alignment designed for training with mixed data (Section 4.3).

## 4.1 Model Architecture

The model comprises three main components. Initially, visual information from the short videos is extracted using a visual encoder. Subsequently, the features are compressed by cross-attention based Q-Former. Finally, they are mapped to the token embedding space of the Large Language Model (LLM) to generate the desired output.

**Visual Encoder**: A pre-trained Vision Transformer (ViT) model equipped with a Global Multi-Head Relational Aggregator is utilized. For a sequence of frames $v$ in a short video clip:

$$v \in R^{T \times H \times W \times C}. \tag{1}$$

Each frame is aprtitioned into $M$ non-overlapping patches, yielding $X_{vis}^t \in R^{M \times d}$. A class token is inserted into each frame along with spatial positional encoding, formally:

$$z_{(0)}^t = [e_{cls}^t, X_{vis}^t] + e^{spa}. \tag{2}$$

The model integrates these features into $L$ stacked ViT blocks $B_{(i)}$. Global Multi-Head Relation Aggregators (GMHRA) are inserted behind specific $K$ ViT blocks $B_{j_0}, ..., B_{j_{k-1}}$ to obtain temporal information across multiple frames. The final step involves concatenating the aggregated features with all frame features to obtain the desired result:

$$z_{(i)}^t = B_i(z_{(i-1)}^t), \forall t \in \{0, ..., T-1\}, i \in \{0, ..., L-1\}, \tag{3}$$

$$z_{gmhra,h+1} = \text{GMHRA}(z_{gmhra,h}, [z_{(j_h)}^0, ...z_{(j_h)}^{T-1}]), h \in \{0, ..., k-1\}, \tag{4}$$

$$Z_{vis} = [z_{gmhra,k}, z^0, ..., z^{T-1}]. \tag{5}$$

**Visual Feature Compression**: The visual features are compressed by a series of learnable queries using the Q-Former structure, as in BLIP2 [13]. These queries are subsequently mapped to the token embedding space compatible with the Large Language Model (LLM) via a linear layer. Within the Q-Former, the learnable queries undergo self-attention to interact amongst themselves and cross-attention to engage with the visual features. Formally, given the visual features $Z_{vis}$ and learnable queries $Q \in R^{n_q \times d}$:

$$Q = \text{QFormer}(Q, Z_{vis}). \tag{6}$$

An alignment layer is introduced to ensure that the outcomes produced by the Q-Former align with the LLM's token embedding space.

$$Q = WQ + b. \tag{7}$$

**LLM Generation**: The decoder employed in our model is Baichuan [40]. Prompt tokens $P_{in}$ are embedded into the input space of LLM, then concatenated with the aligned queries $Q$ to generate the output tokens.

$$T_{out} = \text{LLM}(cat(Q, \text{Emb}(P_{in}))). \tag{8}$$

## 4.2 Prompt-guided Keyword Generation

The most straightforward method to train a fine-grained video description model invloves issuing description prompts, such as "Please describe this video" to the model and expecting it to generate output a desired description for the specified clip. Employing this strategy with our dataset, however, yielded unsatisfactory outcomes. Visual inspection of the results indicated biases in the model's recognition of certain key information. We propose a prompt-guided keyword generation approach to enhance the model's ability to identify key information.

In the data preprocessing phase, key words are extracted from the fine-grained description corresponding to each clip using LLM. During generation, the model receives a description wherein a randomly selected keyword is replaced with placeholders. The model is then tasked with filling in these blanks, compelling it to distill key visual information from the learned queries. Contrary to the description task, where many words may have little relevance to the visual content, this method focuses on crucial visual details. The prompts $P_d, P_k$ and targets $T_d, T_k$ of these two different tasks are illustrated in Figure 5. The training objectives for the description task and the prompt-guided keyword generation task are denoted by $L_d$ and $L_k$, respectively, where

$$L_x = -\sum_{m=1}^{l} \log p(T_{x,m}|Q, P_x, T_{x,1:m-1}), x \in \{d, k\}, \tag{9}$$

$$L = \lambda_d L_d + \lambda_k L_K. \tag{10}$$

## 4.3 Dual Visual Alignment

In the current landscape, short video platforms are inundated with a plethora of advertisement short videos. The exorbitant cost of manual annotation renders the procurement of fine-grained annotations for such an extensive dataset currently impractical. Nonetheless, many publicly available video caption data exist. It is posited that mixed training with lower-quality video captioning data could endow our model with enhanced generalization capabilities. Within this context, data from SVAD is designated as primary data, while general video caption data is classified as auxiliary data.

Due to varying text patterns in different datasets, directly integrating auxiliary data into mixed-dataset training may detrimentally affect the model's performance. To mitigate the negative impact of incorporating auxiliary data, dual visual alignment is used. A dual alignment layer is introduced to project the learnable queries derived from primary and auxiliary data into distinct embedding spaces $Q_{prim}$ and $Q_{aux}$. Specifically,

$$Q_i = W_i\text{QFormer}(Q, Z_{vis,i}) + b_i, i \in \{prim, aux\} \tag{11}$$

Furthermore, two disparate tasks using auxiliary data are explored, encompassing description and prompt-guided keyword generation, as depicted in Figure 5.

## 5 EXPERIMENTS

This section introduces the implementation details, encompassing the dataset utilized, model experimental settings, and evaluation metrics. Subsequently, we delineate our experimental findings, including comparative analyses with other studies, an ablation study, and a text-to-video result as an application.

**Table 4: Comparison experiments with other methods on the SVAD dataset. The LLM-based metrics includes correctness (Cor), detail (Det), temporal accuracy (Tem), and context (Con). The methods marked with * are re-implemented by the authors.**

| Method | Mod | LLM | Conventional Metrics | | | | | LLM-based Metrics | | | | |
|---|---|---|---|---|---|---|---|---|---|---|---|---|
| | | | B@1 ↑ | B@4 ↑ | C ↑ | R ↑ | Mean ↑ | Cor ↑ | Det ↑ | Tem ↑ | Con ↑ | Mean ↑ |
| **General MLLM** | | | | | | | | | | | | |
| BLIP2 [13] | image | OPT-6.7B | 0.9 | 0.1 | 3.5 | 9.6 | 3.5 | 1.64 | 1.30 | 1.31 | 1.94 | 1.30 |
| Qwen-vl [3] | image | Qwen-7B | 16.7 | 2.2 | 7.5 | 16.8 | 10.8 | 1.56 | 2.19 | 1.15 | 1.96 | 1.72 |
| LLaVA(v1.5) [20] | image | Vicuna-7B | 13.9 | 0.8 | 1.1 | 14.3 | 7.5 | 1.61 | 2.30 | 0.98 | 1.94 | 1.71 |
| VideoChat* [14] | video | Vicuna-13B | 27.4 | 6.5 | 21.3 | 26.5 | 20.4 | 1.75 | 2.10 | 1.56 | 2.14 | 1.89 |
| GVT* [5] | video | - | 23.3 | 6.2 | 32.2 | 28.3 | 22.5 | 1.86 | 1.90 | 1.57 | 2.19 | 1.88 |
| Video-LLaVA* [18] | video | Baichuan-13B | 22.3 | 5.9 | 36.8 | 27.4 | 23.1 | 2.27 | 2.37 | 2.09 | 2.17 | 2.23 |
| **Upper Limit** | | | | | | | | | | | | |
| Rewrite | - | - | 39.8 | 10.6 | 99.3 | 41.7 | 47.8 | 4.68 | 3.89 | 4.75 | 4.45 | 4.44 |
| Ours | video | Baichuan-13B | **29.8** | **6.9** | **38.5** | **29.8** | **26.3** | **2.40** | **2.41** | **2.19** | **2.82** | **2.46** |

**Table 5: Ablation study of the proposed methodologies, with PKG denoting prompt-guided keyword generation and Mix representing mixed-dataset training**

| PKG | Mix | B@1 ↑ | B@4 ↑ | C ↑ | R ↑ | Mean |
|---|---|---|---|---|---|---|
| ✗ | ✗ | **30.7** | 6.7 | 28.1 | 27.2 | 23.2 |
| ✓ | | 29.0 | 6.6 | 32.4 | 27.9 | 24.0 |
| | ✓ | 27.9 | 6.8 | 34.3 | 29.0 | 24.5 |
| ✓ | ✓ | 27.4 | **6.9** | **38.5** | **29.8** | **25.7** |

**Table 6: Ablation study on the proportions of prompt-guided keyword generation and description tasks ($\lambda_k$ and $\lambda_d$).**

| $\lambda_k:\lambda_d$ | B@1 ↑ | B@4 ↑ | C ↑ | R ↑ | Mean ↑ |
|---|---|---|---|---|---|
| 0:1 | **30.7** | **6.7** | 28.1 | 27.2 | 23.2 |
| 1:2 | 30.4 | 6.5 | 30.8 | 27.4 | 23.7 |
| 3:2 | 28.8 | 6.4 | 31.7 | 27.3 | 23.6 |
| 5:2 | 29.0 | 6.6 | **32.4** | **27.9** | **24.0** |
| 10:2 | 28.0 | 6.1 | 31.6 | 27.5 | 23.3 |

**Table 7: Ablation study of different experimental settings involving auxiliary data.**

| Dataset Mixed | Task Des | PKG | Method DVA | B@1 ↑ | B@4 ↑ | C ↑ | R ↑ | Mean ↑ |
|---|---|---|---|---|---|---|---|---|
| | ✓ | | | **27.6** | 6.4 | 34.7 | 28.1 | 24.2 |
| ✓ | ✓ | | | 25.2 | 6.2 | 34.3 | 27.8 | 23.5 |
| ✓ | | ✓ | | 20.7 | 5.2 | 33.1 | 27.5 | 21.6 |
| ✓ | | ✓ | ✓ | 26.1 | 6.4 | 36.7 | 28.8 | 24.5 |
| ✓ | ✓ | | ✓ | 25.6 | 6.5 | **38.7** | 29.2 | 25.0 |
| ✓ | ✓ | ✓ | ✓ | 27.4 | **6.9** | 38.5 | **29.8** | **25.7** |

## 5.1 Implementation Details

The dataset comprises one pretrain dataset sampled from the same short video platform as SVAD dataset, and two downstream datasets, SVAD as primary dataset and Youku-mplug caption [37] as auxiliary dataset. Training commences with the pre-training dataset, followed by employing a mixed-dataset approach as detailed in our methodology. The quality of the generated short video descriptions is assessed using four rule-based metrics: BLEU1 [25], BLEU4 [25], CIDEr [32] and ROUGE-L [19] Inspired by Video-ChatGPT [22], We also consider four LLM-based metrics:information correctness, detail orientation, contextual understanding, and temporal understanding. More implementation details can be found in appendix.

## 5.2 Comparative Experiment

Our SVAD-VLM is benchmarked against the state-of-the-art general multimodal large language models (MLLMs) which are evaluated in a zero-shot manner including BLIP2 [13], Qwen-vl [3] and LLaVA(v1.5) [20]. For equitable comparison, we also re-implement a sota video captioning method, GVT [5], and two sota video MLLMs, VideoChat [14] and Video-LLaVA [18]. Additionally, the rewriting of reference annotations by a large language model [2] is evaluated, serving as an approximate upper limit for the SVAD dataset. Quantitative and qualitative outcomes are detailed in Table 4 and illustrated in Figure 6. SVAD-VLM surpasses all competitors across the 10 evaluation metrics outlined in Table 4, demonstrating considerable advantages.

General image MLLMs exhibit suboptimal performance on the SVAD dataset due to its extensive variety, encompassing diverse industries, trending products, and popular games (Figure 3), which might not be represented in the training data of general MLLMs. This diversity makes the SVAD dataset particularly challenging. Furthermore, while general image MLLMs generate video descriptions based on intermediate frame, the Spatiotemporal Fine-grained Video Description (STFVD) necessitates a robust temporal modeling capability owing to the high information density characteristic of short videos.

SVAD-VLM also notably outperforms GVT [5], VideoChat [14] and Video-LLaVA [18]. This superior performance is attributed not to the integration of LLMs but to its tailored designs for short videos.

Given the inherent randomness in the generation process of LLMs, we utilized a large language model, Qwen-72B [2], to rewrite the reference annotations, presenting these evaluations as upper bounds for the metrics. The substantial discrepancy between existing methods and this theoretical upper limit underscores the complexity of STFVD on the SVAD dataset, highlighting the need for further in-depth investigation.

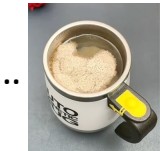 ... 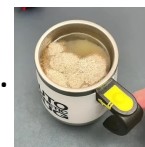 ... 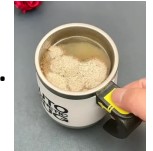 ... 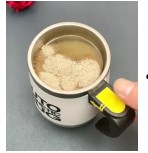 ... 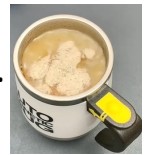 ... 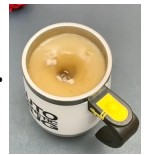 ... 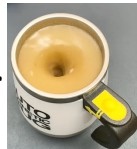

**Reference**: On a table, there is a white automatic blender cup filled with water. A person is pouring a bag of coffee into the blender cup and pressing the yellow button. The blender cup starts stirring automatically.

**BLIP2**: A person is **using a tool** to **clean a cup**.

**Qwen-v1**: The picture shows a multifunctional automatic stirring cup. The cup has a **stainless-steel exterior** with a handle, a white interior, and **stripes** at the top.

**LLaVA (v1.5)**: The picture shows a person holding a cup of coffee, with a **spoon** inside. The cup is placed on a dining table, and **the person is stirring the coffee**.

**VideoChat**: A person is pouring a packet of coffee powder into a white electric stirring cup, then **tapping the lid** on the stirring cup and pressing the button to start stirring. On a **white table** sits a white stirring cup filled with water and coffee powder, then the cup is **tapped open**, and the button is pressed.

**GVT**: On a gray table sits a white stirring cup **filled with** water and **coffee powder**, then a hand presses a button, and the cup starts stirring automatically.

**Video-LLaVA**: A person holds a cup filled with coffee powder, then **uses an electric mixer to stir it**, and **pours the well-stirred coffee into** a white **coffee cup**.

**SVAD-VLM**: On a **table** sits a **white automatic stirring cup**. A person is **pouring a packet of coffee powder** into the cup, then **pressing a button** with their right hand, causing the cup to **automatically stir**.

**Figure 6: Qualitative results from the SVAD dataset, comparing our proposed SVAD-VLM with the BLIP2 [13], Qwen-vl [3], LLaVA(v1.5) [20], VideoChat [14], GVT [5] and Video-LLaVA [18].**

## 5.3 Ablation Study

**The Effect of Proposed Methods**. As depicted in Table 5, significant enhancements in model performance are attributed to both prompt-guided keyword generation and mixed-dataset training techniques. Adopting the two techniques separately brings 15.3% and 22.1% performance improvements to CIDEr, respectively, while adopting them simultaneously improves the performance by 37%. A minor decline in BLEU1 scores primarily results from increased text complexity. This study prioritizes sentence-level similarity to ensure the model accurately extracts key information, rather than achieving word-for-word consistency. Such consistent improvements across metrics affirm the effectiveness of our approach.

**Impact of Task Proportions**. Table 6 presents the effect of varying the ratios of prompt-guided keyword generation tasks to description tasks within the SVAD dataset. The findings suggest that, within a specific range, integrating these two tasks proves advantageous for the model. However, due to the significant difference in output lengths between the keyword generation and the targeted short video descriptions, exceeding this range compromises the model's descriptive capability.

**The Effect of Auxiliary Data Settings**. Through comprehensive experimentation, we explore the effects of diverse experimental setups on the auxiliary dataset, as presented in Table 7. We use post-pretraining as a baseline to illustrate the effectiveness of mixed-dataset training. Post-pretraining indicates the sequence where the model, following its initial pretraining, is trained first on auxiliary data and subsequently on primary data. Direct mixed-dataset training is found to adversely impact the performance by 0.2, 0.4 and 0.3 to BLEU4, CIDEr, and ROUGE-L, highlighting a domain gap between SVAD and other datasets. The introduction of dual visual alignment considerably enhances performance across all evaluation metrics when incorporating auxiliary data. When using the description task and prompt-guided keyword generation task separately on auxiliary data, incorporating dual visual alignment can

lead to performance improvements by 4.8% and 11.1% to BLEU4, 12.8% and 10.9% to CIDEr, as well as 5.0% and 4.7% to ROUGE-L. Furthermore, within the context of auxiliary data, amalgamating the prompt-guided keyword generation task with the description task offers additional benefits, which is by 1.2 compared with description task and by 0.7 with prompt-guided keyword generation task to the average of four metrics. The increase is not as much as on SVAD dataset given that auxiliary data annotations lack the details present in SVAD.

## 5.4 Text-to-video Retrieval

An experiment is conducted on the text-to-video retrieval task utilizing a contrastive learning model similar to CLIP-ViP [38] and results are shown in Table 8. In comparison with general video captions produced by BLIP2 [13], employing spatiotemporal fine-grained video descriptions as text queries significantly improves the recall rate, indicating a closer match to the source video. This outcome further substantiates the efficiency of spatiotemporal fine-grained video descriptions in modeling short videos.

**Table 8: Application on text-to-video retrieval.**

| Text Type | R@1 | R@5 | R@10 | Mean |
|---|---|---|---|---|
| General Caption (BLIP2) | 45.3 | 79.0 | 89.2 | 71.17 |
| Fine-grained Description (Ours) | **86.7** | **99.0** | **99.7** | **95.12** |

## 6 CONCLUSION

We propose spatiotemporal fine-grained video description, a new video-language modeling task provides detailed spatiotemporal descriptions focusing on short video. A new dataset named SVAD has been established to support this research. We suggest adding a completion task during training to efficiently utilize key information in the text. Also, We design a simple yet effective dual alignment layer for mixed-data training to benefit from auxiliary data. Extensive experiments have validated the effectiveness of our approach.

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
