# OpenReview forum: "Spatiotemporal Fine-grained Video Description for Short Videos"
_acmmm.org/ACMMM/2024/Conference — MM2024 Oral_

### Official Review · Reviewer_wjSR · 2024-05-21

**Rating:** 4
**Confidence:** 4

**Summary:**

1. The authors propose the  Spatiotemporal Fine-grained Video Description task and SVAD dataset with 34,930 clips from 5,046 video advertisements.

2. They also introduce SVAD-VLM, a visual-language model adopted prompt-guided keyword generation and dual visual alignment for spatiotemporal fine-grained short video advertisements descriptions.

**Strengths:**

S1: SVAD is the first short video advertisement dataset with fine-grained description annotations.

S2: The SVAD-VLM has a good performance, achieving SOTA on the SVAD dataset. The proposed prompt-guided keyword generation and dual visual alignment are novel and proved useful in the ablation study.

**Limitations:**

W1: The proposed model is primarily compared with LLM-based methods. It would be beneficial to include comparisons with a broader range of non-LLM-based methods.

W2: Although the SVAD dataset includes both English and Chinese, the source videos are entirely from Chinese materials. This may introduce potential biases into the dataset.

W3: The work would benefit from including comparisons with more studies focused on advertising video understanding methods or advertising video datasets, e.g.:

[1] Guo, D., & Zeng, Z. (2021, October). Multi-modal representation learning for video advertisement content structuring. In Proceedings of the 29th ACM International Conference on Multimedia (pp. 4770-4774).

[2] Lin, Q., Pang, N., & Hong, Z. (2021, October). Automated multi-modal video editing for ads video. In Proceedings of the 29th ACM International Conference on Multimedia (pp. 4823-4827).

[3] Tang, Y., Xu, S., Wang, T., Lin, Q., Lu, Q., & Zheng, F. (2022). Multi-modal segment assemblage network for ad video editing with importance-coherence reward. In Proceedings of the Asian Conference on Computer Vision (pp. 3519-3535).

**Suitability:**

3

---

### Official Review · Reviewer_CTTQ · 2024-05-22

**Rating:** 5
**Confidence:** 3

**Summary:**

This work presents a new task: spatiotemporal fine-grained video description, which emphasizes the uniqueness of short videos. This task aims to capture the details of the main subjects and fine-grained movements in short videos. In this paper, a dataset (SVAD) of short videos and detailed descriptions is collected. It consists of 34,930 clip-text pairs covering diverse topics such as products and games. Additionally, a visual-language model (SVAD-VLM) is proposed to generate descriptions for short videos. In this model, a prompt-guided keyword prediction task is involved to compel the model to focus on the key visual information. Extensive experiments are conducted to verify the effectiveness of the proposed model.

**Strengths:**

1. A high-quality dataset of detailed video descriptions is proposed and analyzed.
2. A new visual-text model is proposed for fine-grained short video description.
3. The analysis of the dataset and the experiments is comprehensive.

**Limitations:**

1. Statistics on the duration of the videos in the proposed dataset should be provided.
2. More discussion and comparisons between short videos and long videos would be helpful to validate the motivation of this work.
3. Potential bias in the proposed dataset should be addressed, as there is an obvious frequency gap between words shown in Figure 1 of the supplementary material.
4. More explanations of the implementation details are suggested, such as the training cost in computational resources and time.
5. The limitations of this work should be discussed.

**Suitability:**

3

---

### Official Review · Reviewer_fdF4 · 2024-05-25

**Rating:** 3
**Confidence:** 2

**Summary:**

In this paper, the authors propose the SpatioTemporal Fine-grained Video Description (STFVD) task, which is a more detailed version of the video description task, such as video captioning and dense video captioning. To support this, they created the Short Video Advertisements Description (SVAD) dataset, which includes videos from a wide range of categories along with detailed descriptions. Furthermore, SVAD-VLM is proposed to overcome the difficulty induced by the increased complexity of the new task.

**Strengths:**

(1) The proposed dataset looks highly valuable. Specific descriptions seem to be available for some reasoning tasks as well as for captioning.

(2) It is well written, and examples are well illustrated through the main script and supplementary materials.

**Limitations:**

(1) When I look at the results from the ablation study, I question whether SVAD-VLM is well designed. In fact, as shown in Tables 5 and 7, it can be observed that Bleu@1 performance decreases when the proposed modules are added, even though this metric seems to be the most important compared to other metrics. I believe the authors need to provide an explanation for this.

(2) It is questionable whether it is reasonable to use additional datasets with the proposed alignment technique during benchmarking. It seems necessary to compare the training dataset used with other methods and to provide a justification in the text for the comparison using more data.

(3) The limitations and future directions of this study have not been disclosed. While this is not mandatory and thus has a minor impact on the review, including it would be beneficial. Moreover, for a paper that proposes a new dataset, I believe it is more important to address how the dataset will be managed in the future (e.g., corrections, supplements, disclosures) rather than focusing solely on its initial construction. I am curious to know if there are any plans for this.

**Suitability:**

3

---

### Meta-Review · Area_Chair_MxQF · 2024-07-05

**Recommendation:** Accept (Oral)
**Confidence:** 5

**Metareview:**

The authors propose the SpatioTemporal Fine-grained Video Description (STFVD) task to capture the details of the main subjects and fine-grained movements in short videos. The paper is well written and the strengths include the dataset, novelty, visual language model, and the detailed analysis of the experiments. There were several concerns raised by the reviewers most of which were addressed by the reviewers. Overall the reviewers agree that while there are some limitations the quality of the paper si sufficient to be accepted to the conference.